# ALDH1-Mediated Autophagy Sensitizes Glioblastoma Cells to Ferroptosis

**DOI:** 10.3390/cells11244015

**Published:** 2022-12-12

**Authors:** Yang Wu, Helena Kram, Jens Gempt, Friederike Liesche-Starnecker, Wei Wu, Jürgen Schlegel

**Affiliations:** 1Department of Neuropathology, Institute of Pathology, School of Medicine, Technical University Munich, 81675 Munich, Germany; 2Department of Neurosurgery, Klinikum Rechts der Isar, School of Medicine, Technical University Munich, 81675 Munich, Germany; 3Institute of Pathology and Molecular Diagnostics, Medical Faculty, University of Augsburg, 86156 Augsburg, Germany; 4Department of Radiology, Molecular Imaging Program at Stanford, Stanford University, Stanford, CA 94305, USA

**Keywords:** glioblastoma, cancer stem cells, ferroptosis, autophagy, therapy

## Abstract

The fatal clinical course of human glioblastoma (GBM) despite aggressive adjuvant therapies is due to high rates of recurrent tumor growth driven by tumor cells with stem-cell characteristics (glioma stem cells, GSCs). The aldehyde dehydrogenase 1 (ALDH1) family of enzymes has been shown to be a biomarker for GSCs, and ALDH1 seems to be involved in the biological processes causing therapy resistance. Ferroptosis is a recently discovered cell death mechanism, that depends on iron overload and lipid peroxidation, and it could, therefore, be a potential therapeutic target in various cancer types. Since both ALDH1 and ferroptosis interact with lipid peroxidation (LPO), we aimed to investigate a possible connection between ALDH1 and ferroptosis. Here, we show that RSL3-induced LPO and ferroptotic cell death revealed RSL3-sensitive and -resistant malignant glioma cell lines. Most interestingly, RSL3 sensitivity correlates with ALDH1a3 expression; only high ALDH1a3-expressing cells seem to be sensitive to ferroptosis induction. In accordance, inhibition of ALDH1a3 enzymatic activity by chemical inhibition or genetic knockout protects tumor cells from RSL3-induced ferroptotic cell death. Both RSL-3-dependent binding of ALDH1a3 to LC3B and autophagic downregulation of ferritin could be completely blocked by ALDH inhibition. Therefore, ALDH1a3 seems to be involved in ferroptosis through the essential release of iron by ferritinophagy. Our results also indicate that ferroptosis induction might be a particularly interesting clinical approach for targeting the highly aggressive cell population of GSC.

## 1. Introduction

The reason for the devasting clinical prognosis of human malignant gliomas despite aggressive adjuvant therapies is the high rate of recurrent tumor growth, most likely due to an extremely resistant tumor cell population with stem-cell characteristics [1]. Aldehyde dehydrogenase 1 (ALDH1) has been identified as a biomarker for cancer stem cells (CSCs) in different tumor types including human glioblastomas (GBMs) [2,3]. It has been shown that ALDH1 not only is a marker for this highly resistant cell population, but also contributes to the therapy resistance of GBMs [4].

ALDH1 seems to be involved in detoxification of aldehydes that result from lipid peroxidation (LPO) in glioma cells [5]. High levels of LPO are present in proliferating cells due to high rates of oxidative stress, and these levels dramatically increase due to therapy-induced oxidative stress. Interestingly, ferroptosis also relies on LPO. Ferroptosis is a recently discovered cell death mechanism independent from other forms of cell death including apoptosis or necrosis [6]. So far, no unique biomarker of ferroptosis has been identified, but it has been demonstrated that glutathione peroxidase 4 (GPX4) and the reduction in reactive oxygen species is a central component of ferroptosis. Blockade of GPX4 by RAS-selective lethal (RSL3) or downregulation of glutathione by erastin-induced inhibition of the cystine–glutamate antiporter system Xc leads to this form of iron-dependent cell death [7,8].

The functional interrelationship between these two important regulators in tumor biology could be of interest in glioma CSCs since LPO seems to be functional in both ferroptosis and ALDH1. This could also have clinical implications. Inhibition of ALDH1-dependent resistance against chemotherapy and induction of ferroptosis could theoretically lead to more effective adjuvant therapies. Here, we investigated the functional role of ALDH1 in ferroptosis in human malignant glioma cell lines.

## 2. Materials and Methods

### 2.1. Cell Culture, Reagents, Antibodies, and Plasmid

Human glioblastoma cell lines U87MG (male, 69 years, GBM), LN229 (female, 60 years, GBM), and T98G (male, 61 years, GBM) were purchased from the American Type Culture Collection (ATCC, Manassas, VA, USA) and cultured in DMEM with 10% FBS (Gibco, Dreieich, Germany) under standard cell culture conditions (37 °C, 5% CO_2_). Isolated primary cell lines were cultured in primary cell culture medium, which included 500 mL of RPMI-1640 (Gibco, Dreieich, Germany), 10 mL of B27 (Gibco, Dreieich, Germany), 5 mL of N2 Supplement (Gibco, Dreieich, Germany), 5 mL of nonessential amino acids (Gibco, Dreieich, Germany), 10 mL of L-glutamine (Gibco, Dreieich, Germany), 10 μg of fibroblast growth factor 2 (Peprotech, Hamburg, Germany), 150 ng of transforming growth factor beta 2 (Peprotech, Hamburg, Germany), and 1 μg of epidermal growth factor (Peprotech, Hamburg, Germany). GBM stem-like cell line X01 (female, 68 years, GBM) [9] was cultivated as previously described [10]. Bafilomycin (Sigma, Munich, Germany), deferoxamine (ChemScene, South Brunswick, NJ, USA), RSL3, and ferrostatin-1 (Sigma, Munich, Germany) were dissolved in dimethyl sulfoxide (DMSO) at 10 mmol/L stock concentration (except DFO, 100 mmol/L stock concentration) and stored at −20 °C. The following antibodies were used: anti-ALDH1A3 (ab129815, Berlin, Germany), anti-LC3B (NB100-2220, Abingdon, UK), anti-vinculin (abcam, ab129002), rabbit IgG for immunoprecipitation (7074P2, CST, Leiden, Germany), anti-GFAP (3670, CST, Leiden, the Netherlands), anti-MAP2 (MAB3418, Chemicon, Nürnberg, Germany), anti-Nestin (4760S, CST, Leiden, the Netherlands). Protein G sepharose was from GE Healthcare Life Science (Munich, Germany). ALDH1a3 plasmid (Aldh1a3 (NM_053080) Mouse-Tagged ORF Clone) was obtained from Origene (Herford, Germany).

### 2.2. CRISPR/Cas9 Knockout

LN229 and T98G cells were transfected with CRISPR/Cas9 plasmid pSpCas9(BB)-2A-GFP (PX458) (Addgene plasmid #48138) using ALDH1A3 single-guided RNA as previously described [10].

### 2.3. Glioblastoma Primary Cell Lines

Glioblastoma specimens (IDH wildtype, CNS WHO grade 4 according to the WHO classification of tumors of the central nervous system [11]) were collected from the Department of Neurosurgery, Klinikum rechts der Isar of Technical University of Munich with patients’ informed consent. Collection and use of patient-derived tumor samples were approved by the regional ethics committee. Tissue specimens were washed in PBS containing 1% penicillin–streptomycin and separated into pieces by surgery blades. Tissue pieces was incubated with 5 mL of Papain (with 500 uL DNase) for 30 mins to be isolated into single cells or cell groups. To stop the digestion, 5 mL of Ovocuid (with 500 uL DNase) was added for 5 min. Isolated primary cells were collected and cultured in primary cell culture medium (PDCL#9 female, 86 years; PDCL#13 male, 67 years; PDCL#17 female, 63 years).

### 2.4. Cell Viability Assays

LN229, U87MG, and T98G cells were seeded in 96-well plates (5000 cells/well). Treatment was performed at different concentrations for 24 h, while controls received 0.5% DMSO, only. The proportion of viable cells was determined using the 3-(4,5-dimethylthiazol-2-yl)-2,5-diphenyltetrazolium bromide (MTT, Sigma, Munich, Germany) assay following the manufacturer’s recommendations. Absorbance was examined using an Infinite F200 pro Microplate Absorbance Reader (Tecan, Maennedorf, Switzerland).

### 2.5. LDH Release Assay

For cell death detection, LDH release was quantified using a cytotoxicity detection (LDH) kit (Roche #11644793001, Darmstadt, Germany) and was conducted following the manufacturer’s instructions. Absorbance was examined using an Infinite F200 pro Microplate Absorbance Reader at 490 nm.

### 2.6. Lipid Peroxidation Assays

Cells were seeded on cover slides and treated 24 h before performing the lipid peroxidation assays. Cells were stained by BODIPY^®^ 581/591 C11-Reagent (Thermo Fisher, Bremen, Germany) for 30 min and washed with PBS. Cells were either fixed by 30% PFA to perform fluorescence imaging or trypsinized into single cells to perform fluorescence-activated cell sorting (FACS). Fluorescence images were taken by a Zeiss fluorescence microscope (Zeiss, Munich, Germany) and analyzed using software Image J (National Institutes of Health, Bethesda, MD, USA). FACS results were analyzed using FlowJo (FlowJo Inc., Ashland, OH, USA).

### 2.7. Immunoprecipitation Assays

Glioblastoma cell lines LN229 and T98G were treated and collected after 24 h in lysis buffer (50 mM HEPES, pH 7.5, 10% glycerol, 0.1% Triton X-100, 1 mM dithiothreitol, 150 mM NaCl, 2 mM MgCl_2_, and protease inhibitor cocktail). Then, 30 μg of protein was taken out from the cell lysate as the input sample. Anti-ALDH1a3 (ab129815, abcam, Berlin, Germany) and anti-IgG (7074p2, CST, Leiden, Netherland) were added to the lysate for immunoprecipitation (containing 30 μg of protein), and then incubated at 4 °C overnight. Next, 2 mg of protein-G-Sepharose beads were added to each immunoprecipitation sample for 3 h in 4 °C. After washing, Sepharose beads were collected and diluted with SDS sample buffer (1 M Tris-HCl pH 6.8, SDS, 0.1% Bromophenol Blue, glycerol, and 14.3 M β-mercaptoethanol), and then heated at 100 °C in 5 min. Input samples and immunoprecipitation samples were fractionated by 10% SDS-page gels and read out by Western blotting with anti-LC3b (NB100-2220, Novus bio, Munich, Germany).

### 2.8. Sphere Formation Assays

Glioblastoma established cell lines U87 LN229, and T98G, as well as corresponding ALDH1a3 knockout cell lines, were investigated in sphere formation assays as previously described [10]. Briefly, cells were seeded in ultralow-attachment 12-well plates and cultured under primary cell medium. Pictures were taken after 7 days using a Nikon microscope (Melville, NY, USA).

### 2.9. ALDH1a3 Stable Transfection

ALDH1a3 plasmid was obtained from OriGENE (MG222097). Briefly, PDCL#9 cells (ALDH1a3 negative) were transfected with ALDH1a3 plasmid using lipofectamine 2000 (Invitrogen, Carlsbad, CA, USA). After 48 h, ALDH1a3-overexpressing clones were selected by G418 (Merck, Germany, Munich, 50 mg/mL in H_2_O) treatment for 2 weeks.

### 2.10. Statistics

Three independent experiments of each assay were conducted to validate results. A *t*-test was used for normally distributed data of two unpaired groups. GraphPad Prism 8 (GraphPad Software Inc.; San Diego, CA, USA) was used to perform the analysis, and *p*-values < 0.05 were regarded as statistically significant.

## 3. Results

### 3.1. RSL3-Induced Ferroptosis-Like Cell Death in Glioblastoma Cell Lines

To discover RSL3-induced cell death in glioblastoma, we treated tumor cells of three established glioma cell lines (U87MG, LN229, and T98G), three primary GBM cell lines (PDCL#9, PDCL#13, and PDCL#17), and GSC-enriched glioma cell line X01 with different concentrations of RSL3. We observed the RSL3 dose-dependent loss of cell viability, but it was different between glioblastoma cell lines (Figure 1A). There was an RSL3-resistant group of cell lines (A1G1, PDCL#9, PDCL#13, T98G, and X01) that showed no cell loss at lower concentrations up to 0.8 μM RSL3 and weak effects at higher concentrations (ca. 80% cell viability at 1.6–6.4 μM RSL3). In contrast, a group of RSL3-sensitive cell lines (LN229, U87, and PDCL#17) demonstrated decreased cell viability even at lower concentrations and approximately 30% cell survival at higher concentrations.

The type of RSL-3-induced cell loss was most likely ferroptosis, since concomitant treatment with ferrostatin-1, a specific ferroptosis inhibitor, and with deferoxamine (DFO), an iron chelator, completely abolished RSL3-induced cell death (Figure 1B).

Induction of lipid peroxidation (LPO) after RSL3 treatment was measured using BODIPY staining (Figure 1C). Using this approach, we observed an increase in LPO after RSL3 treatment at 1 μM in all cell lines, which could be completely reverted by concomitant treatment with ferrostatin or DFO (Figure 1D).

These results indicate that the RSL3-induced cell death in glioma cell lines is a ferroptosis-like cell death.

### 3.2. The Sensitivity to RSL3-Induced Ferroptotic Cell Death in Glioblastoma Correlates with the Expression of ALDH1a3

We next verified cell death according to LDH release (Figure 2A). Most interestingly, sensitivity to RSL3-induced ferroptosis correlated with ALDH1a3 expression (Figure 2B). U87, LN229, and PDCL#17 cells exhibited high amounts of ALDH1a3 protein, T98G, PDCL#9, X01, and PDCL#13 showed no expression.

To prove the role of the ALDH1a3 subtype as a glioma stem cell (GSC) marker, we performed sphere formation assays of ALDH1a3 wildtype and corresponding knockout glioma cell lines. Our results showed that ALDH1a3 knockout cells lost their ability to form spheres or formed much smaller spheres compared to their ALDH1a3 wildtype counterparts (Figure 2C).

To further investigate the role of ALDH1a3 in RSL3-induced cell death, we tested the effect of inhibition of ALDH1 enzymatic activity by the specific inhibitor diethylaminobenzaldehyde (DEAB) in ALDH1a3 high-expression (LN229, PDCL#17) and low-expression cell lines (T98G and PDCL#9). DEAB at least partially reversed cell death induced by RSL3 (Figure 3A). These results were corroborated by ALDH1a3 knockout cells that showed similar effects compared with enzymatic inhibition (Figure 3C,D). Moreover, BODIPY staining demonstrated that DEAB treatment also decreased lipid peroxidation induced by RSL3 (Figure 3E,F). In line with these results, ALDH1a3 overexpression after stable transfection in ALDH1a3-negative PDCL#9 cells sensitized the cells to RSL3 treatment, an effect that could be reversed by DEAB treatment (Figure 3G).

These results show that ALDH1a3 seems to be involved in RSL3-induced ferroptotic cell death, and that it might play a role upstream of LPO.

### 3.3. Autophagy Seems to Be Necessary for RSL3-Induced Cell Death

Autophagy is involved in the induction of ferroptosis, and ALDH1 also seems to play a role in autophagy. We, therefore, investigated the effect of autophagy inhibition on ferroptosis in glioma cell lines. The concomitant treatment with the specific autophagy inhibitor bafilomycin A1 rescued glioma cells from RSL3-induced LPO and ferroptotic cell death (Figure 4A–C). Moreover, autophagy by RSL3 treatment was more pronounced in ALDH1a3 wildtype cells than in ALDH1a3 knockout cells (Figure 4D). These results might indicate a role of ALDH1a3 in ferroptosis by autophagy activation. We, therefore, performed a coimmunoprecipitation assay to investigate the connection of ALDH1a3 protein to autophagosomes. After RSL3 treatment ALDH1a3 protein was complexed with LC3, while DEAB treatment eliminated the ALDH1a3–LC3 protein complex formation (Figure 4E). The binding of ALDH1a3 to LC3B-II seems to be relatively weak most likely due to the high autophagic flux in GBM cells and lysosomal degradation [12]. In addition, there seems to be also a binding to LC3B-I. Since it has been shown that ferroptosis depends on the release of iron from ferritin iron storage by autophagy (ferritinophagy), we investigated the ferritin content by Western analysis demonstrating downregulation of ferritin after RSL3 treatment that could be reversed by concomitant DEAB treatment (Figure 4F). These results were corroborated by experiments using ALDH1a3 knockout cell lines (Figure 4G) that showed higher ferritin levels and no decrease after RSL3 treatment.

## 4. Discussion

The fatal clinical course of human glioblastoma despite aggressive adjuvant therapies is due to highly resistant tumor cells, i.e., glioma stem cells (GSC), which express ALDH1 [3], and ALDH1 seems to be involved in the biological processes that lead to therapy resistance by interacting with lipid peroxidation (LPO) [5]. Since ferroptosis is also LPO-dependent [13], we aimed to investigate a possible connection between ALDH1 and ferroptosis.

Here, we show that RSL3-induced ferroptotic cell death revealed RSL3-sensitive and -resistant malignant glioma cell lines. Most interestingly, the RSL3 sensitivity correlated with ALDH1a3 expression; only high-ALDH1a3-expressing cells seemed to be sensitive to ferroptosis induction. In accordance, inhibition of ALDH1a3 enzymatic activity by DEAB protected cells from RSL3-induced cell death.

Ferroptosis is a novel cell death mechanism distinct from necrosis or apoptosis [6]. So far, its role in normal cellular homeostasis is unclear, but it has been shown that ferroptosis induction could offer novel treatment options in aggressive tumor types including GBM [14]. Glioma stem cells (GSCs) characterized by ALDH1a3 expression seem to be the most resistant cell type in GBM, and ALDH1a3 enzymatic activity seems to be involved in the development of the resistant phenotype [10]. Therefore, we expected that combined treatment with an ALDH1 inhibitor (DEAB) and with a ferroptosis inductor (RSL3) should increase cell death in GBM cell lines. However, the contrary happened; ALDH1 inhibition by DEAB nearly completely reverted RSL3-induced ferroptotic cell death. So far, no direct involvement of ALDH1 family enzymes in ferroptosis has been shown.

Since autophagy is involved in both ferroptosis [15] and ALDH1a3 regulation [10], we next investigated the role of autophagy in glioma cell response. We showed previously that, in human GBM, ALDH1 is involved in therapy resistance against the standard chemotherapeutic agent temozolomide by detoxifying toxic aldehydes resulting from oxidative stress and consequent activation of LPO, and that this process seems to be mediated at least in part by autophagy [5]. The results of the present study also support the view that ALDH1 not only is involved in the regulatory processes of autophagy but could be an important regulator or sensor for autophagy. Here, we show that inhibition of ALDH1 enzymatic activity abolished autophagy, and this also was necessary for ferroptotic cell death.

Autophagy plays an important role in ferroptosis [16]. Ferroptosis is an iron-dependent cell death mechanism, and iron is released from ferritin storage by an NCOA4-dependent autophagic mechanism known as ferritinophagy [17]. Therefore, the critical biological process seems to be the release of iron by ferritinophagy. Consequently, ALDH1a3 high-expression cells exhibited high levels of autophagy and ferritin release compared with ALDH1a3 low-expression cells. These data indicate that ALDH1a3 is important in the ferroptosis of human malignant glioma cells by autophagy regulation.

These data are also relevant for clinical consequences, although our results do not support the hypothesis that ferroptosis and concomitant ALDH1 inhibition could increase tumor cell death under standard treatment conditions. However, tumor cells highly expressing ALDH1 represent the most aggressive subpopulation of GSC within GBM. Moreover, it has been shown that the number of ALDH1a3-expressing cells dramatically increases in recurrent tumor growth [18], eventually resulting in the high therapy resistance of recurrences. Our data show that ALDH1a3-positive GSCs that are highly resistant against standard therapy seem to be most susceptible to ferroptosis induction. Thus, ferroptosis could be a therapeutic option in recurrent tumors with high amounts of ALDH1-expressing cells. Therefore, our data might fuel the hope of overcoming resistance to therapy by ferroptosis induction in GBM.

## Figures and Tables

**Figure 1 cells-11-04015-f001:**
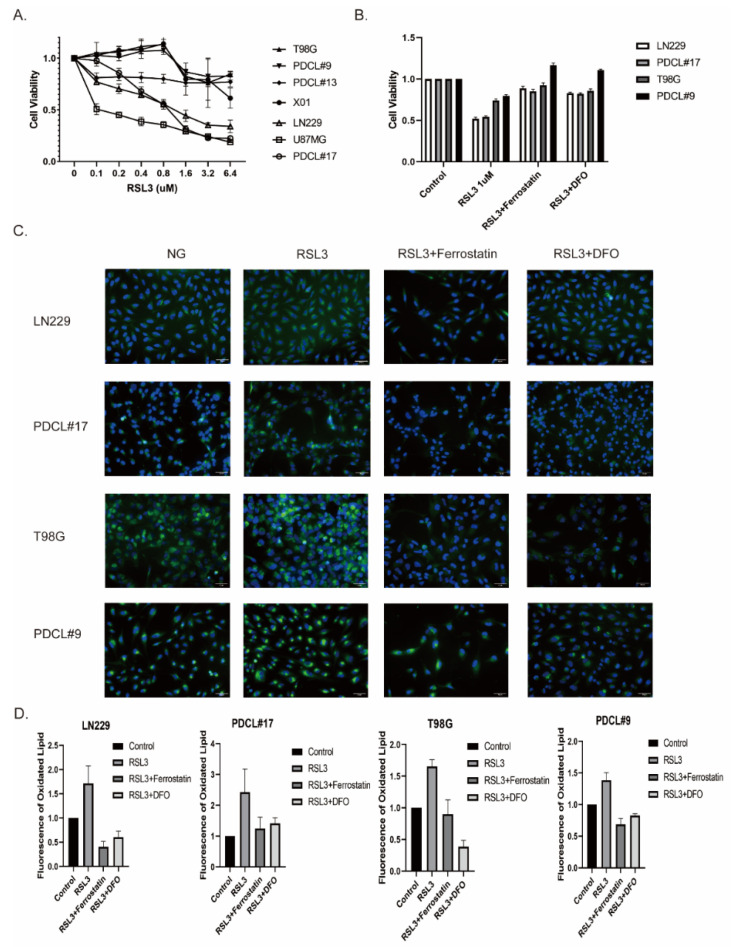
RSL3-induced ferroptosis-like cell death in glioblastoma cell lines. (**A**) Glioblastoma cell lines were treated with increasing doses of RSL3 (0 μM, 0.1 μM, 0.2 μM, 0.4 μM, 0.8 μM, 1.6 μM, 3.2 μM, and 6.4 μM), cell viability was assessed by MTT assays. Cell lines A1G1, T98G, PDCL#9, PDCL#13, and X01 showed only a mild reduction in cell viability, whereas LN229, U87MG, and PDCL#17 exhibited a significant reduction. (**B**) RSL-3-sensitive LN229 and PDCL#17 cells, and RSL-3 resistant T98G and PDCL#9 cells were treated with RSL3 combined with ferroptosis inhibitor ferrostatin and iron chelation DFO; cell viability was determined by MTT assays. Both ferrostatin and DFO completely abolished the RSL-3 induced cell death. (**C**,**D**) RSL-3-sensitive LN229 and PDCL#17 cells, and RSL-3-resistant T98G and PDCL#9 cells were treated with RSL3, combined with ferrostatin and DFO, respectively; lipid peroxidation (LPO) induced by RSL3 treatment was determined using BODIPY C11-581/591 reagent, as shown by green fluorescence. Results are quantified in (**D**) (*p*-values: untreated control vs. RSL3 < 0.005, RSL3 vs. RSL3 + ferrostatin < 0.005, RSL3 vs. RSL3 + DFO < 0.005). All cell lines showed pronounced induction of LPO that could be completely blocked by ferroptosis inhibitors.

**Figure 2 cells-11-04015-f002:**
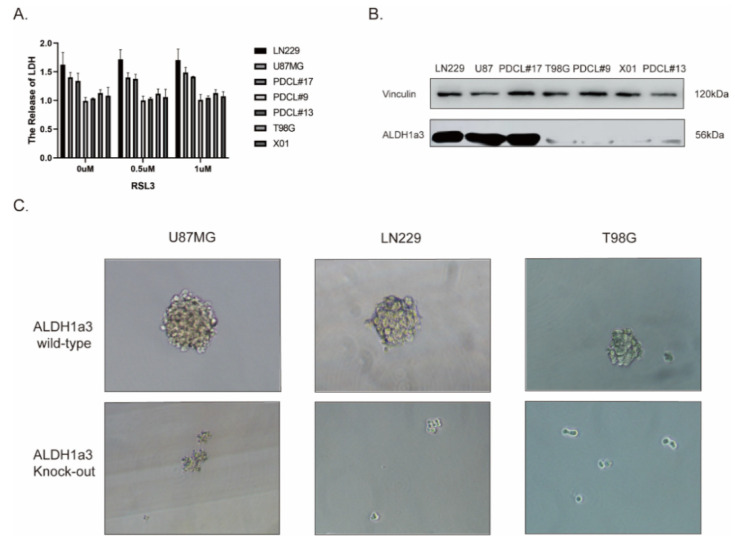
The sensitivity to RSL3-induced ferroptotic cell death in glioblastoma correlates with expression of ALDH1a3. (**A**) Relative LDH release (1.0 = untreated controls) after RSL3 treatment confirmed the cell viability results. RSL3 induced cell death in LN229, U87MG, and PDCL#17 cells, but not in in T98G, PDCL#9, PDCL#13, and X01 cell lines. (**B**) Expression of ALDH1a3 protein was assessed in seven cell lines by Western blot analysis. RSL-3-sensitive LN229, U87MG, and PDCL#17 cells showed strong expression of ALDH1a3, whereas RSL3-resistant T98G, PDCL#9, PDCL#13, and X01 cells showed only weak expression. Vinculin served as a loading control. (**C**) ALDH1a3 high-expression U87MG and LN229 cells, and ALDH1a3 low-expression T98G cells, respectively, and their corresponding ALDH1a3 knockout cells were seeded into ultralow-attachment plates and were grown under primary cell culture conditions for 7 days. All three cell lines showed sphere formation with much smaller spheres in T98G cells under primary cell culture conditions. Sphere formation was completely abolished in knockout cells.

**Figure 3 cells-11-04015-f003:**
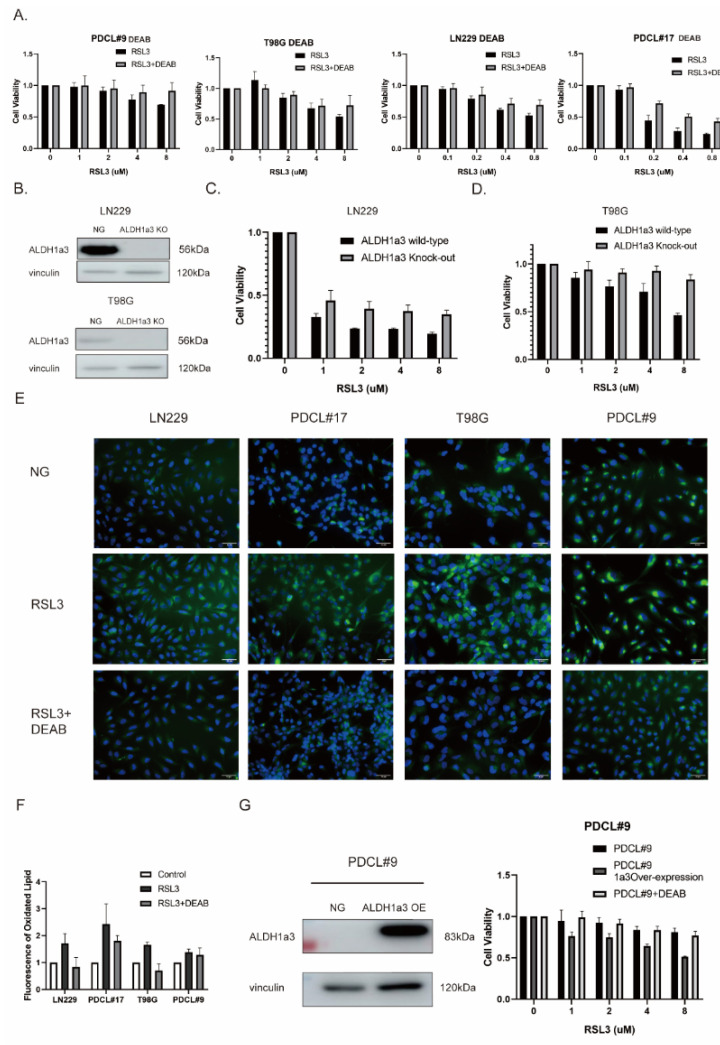
(**A**) RSL3 treatment with and without DEAB in T98G and PDCL#9 (0 μM, 1 μM, 2 μM, 4 μM, and 8 μM), and RSL-3 sensitive LN229 and PDCL#17 cells (0 μM, 0.1 μM, 0.2 μM, 0.4 μM, and 0.8 μM); cell viability was determined by MTT assays. All cell lines showed decreased cell viability (RSL-3-resistant cells at 10-fold higher concentrations), which could be partially reverted by ALDH1a3 inhibition using DEAB (*p*-values: PDCL#9 8 μM < 0.05; T98G 8 μM < 0.05; LN229 0.8 μM < 0.05; PDCL#17 0.2 μM < 0.01, 0.4 μM < 0.01, 0.8 μM < 0.05). (**B**) Western blot analysis in LN229 and T98G, demonstrating the results of ALDH1a3 knockout. (NG, negative control; KO, knockout). (**C**,**D**) RSL3 treatment (0 μM, 1 μM, 2 μM, 4 μM, and 8 μM) in RSL-3 sensitive LN229 and T98G ALDH1a3 wildtype and knockout cells showed partial reduction in cell death in ALDH1a3 knockout cells (*p*-values: LN229 each group < 0.05; T98G 4 μM < 0.05, 8 μM < 0.005). (**E**) BODIPY 581/591 C11 staining of lipid peroxidation after RSL3 with or without DEAB treatment demonstrated a significant reduction in LPO after concomitant DEAB treatment. (**F**) Quantification of BODIPY 581/591 C11 fluorescence staining (*p*-value: LN229 control vs. RSL3 < 0.005, RSL3 vs. RSL3 + DEAB < 0.05; PDCL#17 control vs. RSL3 < 0.05; T98G control vs. RSL3 < 0.001, RSL3 vs. RSL3 + DEAB < 0.05; PDCL#9 control vs. RSL# < 0.001). (**G**) Protein expression after stable ALDH1a3 transfection in ALDH1a3-negative PDCL#9 cells demonstrated strong ALDH1a3 overexpression in transfected cells. Cell viability was measured after RSL3 treatment with or without DEAB by MTT assays. ALDH1a3-overexpressing PDCL#9 cells demonstrated a reduction in cell viability after RSL-3 that could be reverted by DEAB treatment (*p*-values: PDCL#9 vs. PDCL#9-A3 OE in 4 μM < 0.001, in 8 μM < 0.001; PDCL#9A3-OE vs. PDCL#9A3-OE + DEAB in 1 μM < 0.05, in 2 μM < 0.05, in 4 μM < 0.01, in 8 μM < 0.01).

**Figure 4 cells-11-04015-f004:**
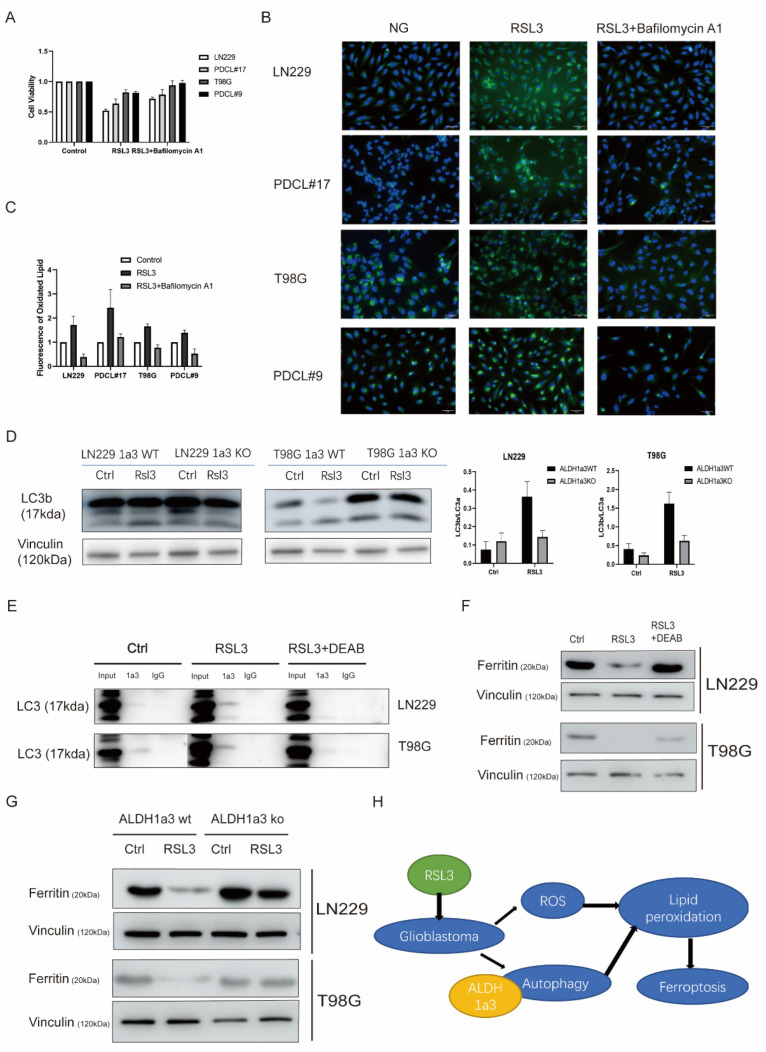
Autophagy in RSL3-induced cell death. (**A**) Cell viability after RSL3 with or without Bafilomycin A1 in LN229, PDCL#17, T98G, and PDCL#9 showed reversion of cell viability by autophagy inhibition by bafilomycin A1 (*p*-value: LN229 RSL3 vs. RSL3 + Bafilomycin A1 < 0.01; PDCL#17 RSL3 vs. RSL3 + Bafilomycin A1 < 0.01). (**B**) BODIPY 581/591 C11 staining of lipid peroxidation after RSL3 treatment with or without Bafilomycin A1 showed a significant reduction in fluorescence staining after autophagy inhibition. (**C**) Quantification of lipid peroxidation after RSL3 with or without Bafilomycin A1 treatment. (**D**) Western blot analysis and quantification of LC3B/LC3A ratio of ALDH1a3 wildtype and knockout cells showed a pronounced induction of autophagy in wildtype cells but not in ALDH1a3 knockout cells. Vinculin served as control. (**E**) Immunoprecipitation assays of LN229 and T98G cells; protein complexes were precipitated with ALDH1a3 antibodies after RSL3 treatment with or without DEAB. Immunoblots showed ALDH1a3 binding to LC3 after RSL3 treatment that could be reverted by concomitant DEAB treatment. (**F**) Western blot analysis of ferritin expression after RSL3 treatment with and without DEAB in LN229 and T98G cells. Ferritin iron storage was downregulated after RSL3 treatment, which could be reversed by concomitant DEAB treatment. Vinculin served as control. (**G**) Western blot analysis of ferritin expression after RSL3 treatment in LN229 and T98G ALDH1a3 wildtype or ALDH1a3 knockout cells. The results corroborated the results of (F), showing ferritin downregulation after RSL-3 treatment only in ALDH1a3 wildtype cells. (**H**) Schematic diagram: RSL3 treatment induces ROS accumulation and activates ALDH1a3-dependent autophagy in glioblastoma cells, leading to lipid peroxidation and finally to ferroptosis.

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
