# Peer review of "ALDH1-Mediated Autophagy Sensitizes Glioblastoma Cells to Ferroptosis"

_cells, 2022, doi:10.3390/cells11244015_

Round 1

Reviewer 1 Report

In the manuscript, the authors claim that ALDH1a3 regulates autophagy by binding to LC3B and modulates cell susceptibility to ferroptosis. In its current state, I'm afraid the data on the binding of ALDH1a3 to LC3B do not adequately support the conclusion. Although the data is potentially interesting, I have several points that should be addressed by the authors. Specific comments are as below.

Major points.

1.      The results in Figure 1 show that sensitivity to RSL3 varies widely among glioblastoma cell lines, with some cells being more effective and others less effective. The authors claim that ALDH1a3 is involved in ferroptosis sensitivity, but they do not provide a correlation between the expression levels and the sensitivity in the cell lines they used (A1G1, T98G, PDCL#9, #13, X01, LN229…). The authors should perform western blotting and/or RT-PCR and clarify this point.

2.      Overall, MTT assay is a cell proliferation assay and does not quantify cell death itself. The authors should perform a bona fide cell death assay, such as propidium iodide staining or LDH release assay instead.

3.      The efficiency of knockout of ALDH1a3 in the cells should be confirmed by western blotting in order to show that ALDH1a3 is knocked out in the cells (e.g. Figure 2A, 3G…).

4.      Figure 3E is an important basis for the authors to claim that ALDH1a3 regulates autophagy by binding to LC3B. Technically speaking, it would be preferable to swap the lanes of ALDH1a3 antibody and IgG (negative control) because of concerns about the input samples leaking into adjacent lanes. Also, the results appear to show binding to ALDH1a3 and LC3-I, but less binding to LC3-II (autophagosome membrane-associated form). This point should be discussed.

Minor points.

1.      Scale bars should be indicated all the images (e.g. Figure 1C, 2A, 2E…).

2.      The authors should provide the raw data of western blotting as a supplementary file.

3.      English should be carefully revised by a native English speaker or a professional language editing service.

Reviewer 2 Report

This paper entitled “ALDH1-mediated autophagy sensitizes glioblastoma cells to ferroptosis” by Yang Wu et al. investigated the connection between ALDH1 and ferroptosis in malignant glioma cell lines. Apart from that, the authors indicated that RSL3-induced LPO and ferroptosis in malignant glioma cell lines.   

Comments:

1.     The authors should briefly characterize the cell lines studied.

2.     The manuscript lacks a clear statement of the specific hypothesis. 

3.     Limitation of this study should be further discussed.

4.   The quality of the figures is poor and should be improved. Words are too small to read.

5.  The photos are missing the scale bars.

Round 2

Reviewer 1 Report

Response no. 4 is insufficient in terms of content. Please read my comment carefully and comment sincerely.

Author Response

Dear Reviewer:

We appreciate your considerate comment. 

Your comment is correct. We now added a comment on the binding of ALDH1a3 to LC3B into the Results section of the manuscript (3.3.).

We did not adequately answer the question before for several reasons:

The main point is that we now formulated our hypothesis more precisely following the reviewers‘ comments. The central point is that ALDH1a3 seems to be involved in ferroptosis by its involvement in autophagy leading to iron release by ferritinophagy. We, therefore, investigated the ferritin content by Western analysis demonstrating down-regulation of ferritin after RSL3 treatment that could be reversed by concomitant DEAB treatment (Figure 4F).

We do not claim that the regulation is mediated by binding of LC3B. It just shows that ALDH1a3 is bound to autophagosomes. We hope, that we now were able to explain this aspect more precisely.

The technical aspects are another point of critics.  Leaking into adjacent lanes could be a problem in Western blotting. However, we are confident that this did not occur in our experiments. Moreover, we have shown in our previous papers with other techniques including proximity ligation and IP with p62 that ALDH1a3 is involved in autophagy (Wu, W., et al. 2018. Cancer Lett 417: 112-123. doi: 10.1016/j.canlet.2017.12.036; Wu, W.et al. 2020. Transl Oncol 13: 100748. doi: 10.1016/j.tranon.2020.100748.; Wuerstle, S., et al. 2017. Oncol Lett 14: 322-328. doi: 10.3892/ol_xxxxxxxx).

The reason for the weak LC3B-II signal is most likely due to the high autophagic turnover in GBM cell lines and lysosomal degradation. This aspect has been described by other authors and has also been included in the guidelines for the use and interpretation of assays for monitoring autophagy.

In addition, the time window for revision given by the editor was very narrow and some of the antibodies have delivery restrictions in Germany. Therefore, we decided not to repeat the experiment. We hope, that the reviewer accepts our comments.

Reviewer 2 Report

The authors responded to all the reviewer's comments.  

Author Response

Dear Reviewer:

Thank you very much for your comments! Your suggestions have enabled us to improve our work.

Round 3

Reviewer 1 Report

The authors have addressed almost all of my concerns and I have no further comments.